# The Impact of Long-Term Care Needs on the Socioeconomic Deprivation of Older People and Their Families: Results from Mixed-Methods Scoping Review

**DOI:** 10.3390/healthcare11182593

**Published:** 2023-09-20

**Authors:** Georgia Casanova, Rossella Martarelli, Francesco Belletti, Carolina Moreno-Castro, Giovanni Lamura

**Affiliations:** 1IRCCS-INRCA National Institute of Health & Science on Ageing, Centre for Socio-Economic Research on Ageing, 60124 Ancona, Italy; r.martarelli@inrca.it (R.M.); g.lamura@inrca.it (G.L.); 2International Center for Family Studies (CISF), 20122 Milan, Italy; francesco.belletti@stpauls.it; 3Research Institute on Social Welfare Policy (POLIBIENESTAR), University of Valencia, 46022 Valencia, Spain; carolina.moreno@uv.es

**Keywords:** aging, older people, long-term care, poverty risk, household, review

## Abstract

Background: Long-term care (LTC), poverty, and socioeconomic deprivation are globally significant social issues. Ongoing population aging trends and the recent social and health emergencies caused by the COVID-19 pandemic crisis have highlighted the need for macro-level LTC and welfare system sustainability strategies. Aims: This scoping review (ScR) explores the relationship between LTC needs, the health status of older people, and the risk of socioeconomic deprivation for their households. Methods: The methodology considers different relevant sources: (a) the guidelines for ScR proposed by Lockwood et al.; (b) the recommendations of Munn et al.; (c) the PRISMA guideline for scoping reviews; and (d) the Joanna Briggs Institute (JBI) checklist. Sixty-three papers are included in the mixed-methods analysis. Results: The findings reveal the existence of a debate that seeks to understand the different characteristics of the relationship between the investigated issues. Relevant gaps in the literature are identified in terms of the concepts and approaches of the studies analyzed. Conclusions: The results indicate that the reciprocal relationship between LTC needs, supply, and the risk of socioeconomic deprivation is understudied. Future studies should focus on the causal relationship between the two phenomena and identify any internal factors that may be involved.

## 1. Introduction

In recent decades, the literature has revised the concept of poverty, which was traditionally defined in terms of income level [1], offering a vision of poverty as a more complex, articulated, and multidimensional phenomenon [2,3] that is characterized by an intrinsic interconnectedness between different dimensions, including economic, social, and human opportunities (e.g., school and health system accessibility, job availability, households structures, territorial availability of resources, and accessibility to services) [4]. This is well-reflected in the international plans developed to counteract multidimensional poverty, such as those identified by initiatives including “Transforming our world, the 2030 Agenda for sustainable development” and the “Third ten-year action plan for the eradication of poverty (2018–2027)” [5], which promote the dissemination of studies for a more in-depth understanding of the dimensions of deprivation in order to target better those population segments characterized by specific social needs, for instance, those related to long-term care (LTC) conditions.

The impact of population aging on health and welfare systems around the world is widely recognized [6,7,8], resulting in an increase in the demand for formal and informal care [9] and making LTC a priority for national and international policies [10,11,12,13,14]. In this regard, European LTC schemes are complex combinations of health and social policies, services, and interventions [6,15], whose sustainability is threatened by demographic and fiscal circumstances [16] and, to an even greater extent, by the recent COVID-19 pandemic. In this context, reducing inequalities in health and LTC provision remains a central pillar for many countries’ sustainable development [17,18].

Previous studies underlined the higher risk of social exclusion and social inequality for informal carers, who are often women who frequently feel compelled to limit their work and social lives to care for their relatives [19]. Over and above the indirect cost of LTC provision, out-of-pocket expenditure for private care is rising, even in advanced social protection systems [20]. For these reasons, Mitra and colleagues have recommended that future research should focus on the private side of LTC expenditure borne by families [21]. Within this framework, several studies have investigated and found that older people living in materially deprived conditions have a diminished ability to cover their own care needs [22,23], a situation that has a significant impact on both their psychosocial well-being [24,25] and cognitive health [26]. Despite these efforts, the literature largely overlooks the effects of health conditions on the socioeconomic status and related risk of socioeconomic deprivation (SED) of either dependent older people or the family members who care for them. Similarly, at the policy level, initiatives and schemes supporting family carers do not seem to underpin these situations fully and are, therefore, unable to adequately counteract the risk of poverty and social exclusion arising from informal care activities for dependent people [27]. In light of the current state of affairs, there is an urgent need for a greater focus on the relationship between LTC needs and the risk of socioeconomic deprivation and poverty to understand better the dynamics underlying this phenomenon and how innovative policies can be formulated globally to tackle it. This scoping review study (ScR) seeks to contribute to the debate on this specific issue, thereby supporting future research on how health-related LTC expenses affect the financial situation of care recipients and the family members who care for them. Specifically, this study identifies the primary research gaps and examines how the scientific literature addresses the multidimensional perspective of the socioeconomic deprivation concept.

This study is conducted within the framework of the Family International Monitor (FIM) and the “Socio Economic deprivation related to effect of the presence of Dependent older people: strategies for Innovative Policies in Europe SEreDIPE project (Horizon 2020 MSCA-IF-2019 Grant Agreement No. 888102). Using a multidimensional perspective of the concepts of “family” and “deprivation” [28], both projects are concerned with familial material and social deprivation, with a particular focus on care needs.

## 2. Materials and Methods

To ensure the highest possible standards of reporting, this ScR is based on a methodology that considers the recommendations formulated by the following relevant sources: (a) the guidelines for ScR proposed by Lockwood et al. [29]; (b) the Munn et al. [30] recommendations; (c) the PRISMA guideline for scoping reviews [31]; and (d) the Joanna Briggs Institute (JBI) checklist [32]. The chosen guidelines were coherent and non-overlapping, as possible risks (e.g., Lockwood, including suggestions from PRISMA guidelines and the JBI checklist) were adequately considered. The full details of this study protocol are described in Martarelli et al. [33]. Combining these methods ensured that the review’s path remained linear and focused, according to Lockwood and Munn’s recommendations. At the same time, the PRISMA and JBI approaches concurrently limited the loss of potentially valuable papers on the topic.

Moreover, specific guidelines supported different aspects, such as the suitability of the chosen methods (JBI checklist) and the analysis of data (PRISMA). Lastly, incorporating these suggestions enabled the authors to consider the pre-planning phase as the starting point for the design of the ScR study protocol. This allowed the authors to focus on a complex and multidimensional issue, such as the relationship between LTC needs and care strategies and the risk of SED. Figure 1 depicts the ScR’s flowchart.

### 2.1. Pre-Planning

Lockwood and colleagues [29] pointed out that pre-planning was the phase that determined a review project’s success. The brainstorming and brief preliminary research conducted during this phase enabled the authors to clarify the conceptual framework, determine specific research questions, and identify the set of keywords necessary to implement the search.

### 2.2. Conceptual Framework

The relationship between LTC care needs and SED risk is composed of three main elements: (a) care needs, often expressed through the identification of a specific target of study; (b) socioeconomic deprivation, understood as a multidimensional factor; and (c) the characteristics of the relationship between these two factors.

Figure 2 illustrates that there are two possible directions in which this relationship can develop. The first relates to the situation of people, including those in later life, who live in SED conditions and can therefore count on the reduced availability of social, health, and economic resources [22,23], which in turn contributes to a diminished self-care capacity, as well as the deterioration of their health, autonomy, and overall living conditions [23,24,25,26,27,28,33]. The other direction concerns dependent older people with a reduced self-care ability, who seek to cover their LTC needs via healthcare-seeking behaviors based on cost-coping mechanisms, such as the direct buying of care provision [34,35] or via informal care (e.g., a reduction in employment income) [36,37]. Independently of personal economic conditions and welfare state characteristics of the country, these mechanisms impact directly or indirectly (e.g., by taxation rate) on the socioeconomic status and, consequently, the associated SED risks for both older people and their family caregivers (co-residing or otherwise) [38,39]. To analyze these mechanisms, this study used the concept of multidimensional deprivation based on Erikson’s theory [40], as it allowed us to emphasize that SED encompassed more than just material deprivation and economic impoverishment and to underline that economic and social inclusion aspects were core dimensions to take into account when examining the effects of care strategies for dependent older people and family caregivers.

### 2.3. Research Questions, Methods, and Keyword Identification

This review examined the scientific literature to explore the relationship between LTC needs and the risk of socioeconomic deprivation for older people and their caregiving relatives. At the end of the pre-planning phase, three specific research questions were formulated to address this general objective: (1) to scan the literature on the topic of older adults who required LTC and their socioeconomic status; (2) to identify any conceptual gaps and the most debated unresolved issues in the literature; and (3) to determine the extent to which the so-called “multidimensional perspective” was applied to the SED concept. To this end, the authors chose a mixed quantitative and qualitative approach based on frequency and content analysis. Table 1 shows an overview of the analytical categories considered concerning the three research questions and relative tables in the Section 3.

As shown in Figure 2, the authors identified a set of keywords to cover the chosen conceptual framework’s concepts and relationships. As detailed in the protocol paper [33], the authors searched various databases using the keywords defined in the pre-planning phase that were strictly related to the abovementioned objective. Thirteen keywords were included in the first set of searches: “long-term care”, “older people”, “elderly”, “aged”, “caregiver(s)”, “family caregiving”, “impoverishment”, “deprivation”, “socioeconomic deprivation”, “economic”, “economic impact”, “poverty”, and “multidimensional poverty”. After the initial exploratory searches, additional keywords were added progressively in order to refine the search: “household”, “expenditure”, “healthcare expenditure”, “spending”, “payments”, “economic impoverishment”, “costs”, “burden”, “socioeconomic status”, “socioeconomic/socioeconomic”, “household”, “social differences”, “informal care”, “care”, “carers”, “(inter)generational”, “activities of daily living”, “ADL limitations”, “functional limitations”, “disability”, “life expectancy”, “health”, “health problems”, “income”, “low-income”, and “low-income countries”. Forty-one keywords were used since they were deemed congruent with the conceptual framework (Figure 2).

### 2.4. Selection Process

The entire search process was conducted between March 2021 and April 2022. Four crucial research databases were accessed: Pubmed, Scopus, Web of Science, and Wiley Online Library. A few items were also extracted from non-digital archives or other electronic databases, i.e., “Journal Storage” (JSTOR) and “Cambridge Core” (the books and journals platform from Cambridge University Press). As indicated above, all of the selected search terms were English words. Figure 3 details the search strings used in the different search engines.

As a result of the 24 different keyword combinations emerging from the search process (see [33] for details), 21,200 items, excluding duplicates, met the criteria for selection. They were screened for the scoping review, i.e., included or excluded according to the study protocol’s criteria.

The following articles were chosen for inclusion on the basis of these selection criteria: (a) those focused on the relationship between poor health and the aging process, long-term care needs, and the socioeconomic deprivation of chronically ill older people and their families; (b) those proposing solutions to the economic problems triggered by health needs; (c) those proposing social innovation policies; (d) those based on a specific method (quantitative or qualitative) or mixed methods (i.e., either of these categories); (e) both surveys and systematic or scoping reviews; (f) those referring to “primary” or “secondary” studies; (g) those conducted in high-income or low- and middle-income countries (i.e., either one of the latter two; articles based on a comparative perspective were also included); (h) those that were published within the past five years (exceptions to this rule were articles chosen due to the relevance of the sources, published within the past ten years as a maximum); (i) those written in English; and (j) those published in peer-reviewed journals. Two researchers (GC and RM) independently screened the extracted items based on the titles and abstracts. Ultimately, 21,131 articles were excluded for failing to meet the criteria. Therefore, a total of 69 articles were provisionally selected. A second check of excluded and included papers was undertaken, including a total of 63 papers in the ScR. No other references were found by manual searching or by analyzing the references of included articles. Appendix A contains the complete list of selected papers, including their bibliographic data.

### 2.5. Data Extraction

In order to organize the information for analysis purposes, the authors arranged the collected papers by date, from oldest to newest, then numbered and labeled them sequentially from 1 to 63. Based on a modified JBI data extraction form, a set of 9 analysis categories were determined per the ScR goals and typologies of analysis (Table 1). Two researchers (GC and RM) independently extracted the items based on the identified categories. To collect common information, a thematic and content analysis [41] based on the ex-post categorization of variables [42,43] was performed to (1) detect the presence of variables in each selected study and (2) identify the selected variable’s different modalities. Moreover, a specific dataset was realized to collect the qualitative data to detect: (1) the main content results on the relationship between LTC needs, the health status of older people, and SE conditions of families; (2) identify suggestions for future studies and insights for policymakers; and (3) make them easy to read based on the classification and summarization of specific contents. A summarized table of the content data collected is detailed in Table A2.

### 2.6. Data Analysis and Reporting

The quantitative analysis was based on the frequency calculation of internally determined modalities for each selected category and summarized in reporting tables (Table 2, Table 3, Table 4, Table 5 and Table 6). Given their complexity, additional details were provided for three of the variables in order to clarify their internal definitions better. First, 11 different modalities were identified based on the nine dimensions used by Erikson’s theory to measure the multidimensionality of the deprivation concept utilized by the selected studies. The authors decided to separate “material state” from “network ties” and “social integration” for a better correspondence with the dimensions utilized in the articles and to provide a more accurate evaluation of the deprivation concept’s multidimensional degree. The final list of dimensions is detailed in Table 4. Secondly, the degree of multidimensional deprivation was calculated by adding the number of dimensions used by each article. The definition of three multidimensional levels (low, medium, and high) facilitated the observation of the distribution of levels in the deprivation’s multidimensional concept. Lastly, the World Bank classifications of the country’s income level (low, medium–low, medium–high, and high) were applied and are reported in Table 5. The qualitative part of the study used a descriptive interpretative approach to provide an in-depth understanding of the contents of reviewed papers. After an in-depth reading of the reviewed articles, two authors (RM, GC) identified 22 papers relevant to their contents. According to the explorative aim of the scoping review strategy, the two authors used thematic and content analyses in this selection of papers to better focus the qualitative analysis on significant results.

## 3. Results

The ScR found 63 papers in the ten years covered. This study’s first finding was that there was a certain level of interest in the scientific literature regarding the association between older people’s health and socioeconomic conditions. The quantitative results are summarized in Section 3.1 according to the selection of variables identified in Table 1 for the quantitative analysis. Section 3.2 refers to qualitative results from selecting 22 papers identified for other content relevance by the authors, as explained in the Section 2 Materials and Methods.

### 3.1. Descriptive Quantitative Results

#### 3.1.1. LTC Needs Defined by Targets: Older People, Caregivers, and Households

As for the relationship between people’s LTC needs and deprivation dimensions, 80% of the analyzed articles targeted a specific population (Table 2). Specifically, older people were the most researched target (23 of 63 articles), followed by households (15 articles; 23.8%) and caregivers (13 articles; 20.6%).

The in-depth analysis of the data reported in Table 2 confirms the prevalent research strategy of targeting older people by mixing the groups of the oldest old (80 years or older) and younger senior population (65–75 years old) in order to estimate the potential level of care needs.

A case in point was provided by Flores-Flores et al. [44], who focused on the impact of poverty on health insurance opportunities and the use of preventive services. Their study included three different age groups: 65–70, 71–75, and 76–80 years old. The study also showed a higher incidence of limitations in activities of daily living among the oldest old, whose rate of disability was about five times that of people aged 36 to 64 years. The study by Wilkinson. et al. [45] also offered a clear example, as it targeted Medicare beneficiaries aged 65+ years to emphasize their needs for all the services that Medicare, the well-known federal health insurance program in the USA, does not cover (i.e., long-term services and support for personal care and assistive devices). This article investigated the extent to which the financial burden of older Americans was commensurate with the level and intensity of their care needs.

Moreover, some studies applied a different concept of “older age” due to the need to investigate not only the age group to which an individual belongs, but also whether or not the average age at first infirmities tended to change significantly over time. They not only looked into how old “older people” were, but also the age at which older adults were “really old”. To this end, they covered a broad spectrum of individuals aged 60 years and older. For example, Murayama et al. [46] conducted a study on the long-term changes in a functional capacity among older people in Japan (2020). Based on the data drawn from the National Survey of the Japanese Elderly (NSJE), this study focused solely on those aged 60 years and older at baseline. The Myanmar Aging Survey (MAS) also used a sample of persons aged 60 years and older, as described by Teerawichit Chain et al. [47]. Their article defined “older people with long-term care needs” as those reporting one or more physical difficulties, not only the inability to perform activities of daily living—both instrumental and non-instrumental activities, i.e., IADL and ADL, respectively—but also difficulties with physical functions, such as “lifting 5 kg in weight”, “walking up and down stairs”, “walking 200 to 300 m”, “crouching/squatting”, and “using fingers to hold things”.

The second-largest category of studies, comprising nearly a quarter of the 63 papers analyzed, concerned those who saw the household or the head of the household as their main research target. In this case, the research focused on the relationship between the health conditions of older family members and eventual material deprivation aspects for a specific member (e.g., an older member, head of the household) or the entire family. An example of this approach was provided by Guerchet et al. [48], whose investigation focused on how the presence of care-dependent older members affected the economic functioning of their households, classified according to disease evolution and level of persistence (for instance, by distinguishing between “chronic-care households” and “incident-care households”). This 2018 study was characterized by its use of reliable financial strain indicators (e.g., loans, shares, and extra work) and examination of a wide range of household income components (both stable and transitory). The article by Salari et al. [49] on the most relevant household characteristics associated with “catastrophic health payments” is another example in this regard. Based on the data from the Kenya Household Health Expenditure and Utilization Survey 2018 (KHHEUS), this study concluded the impoverishment effect of the presence of older members, particularly regarding the health-seeking behavior of those afflicted with chronic diseases. In addition, Zhao et al. [50] investigated the caregivers as the study’s primary research target, explicitly focusing on informal care contexts and the implications on caregivers’ quality of life and social and material deprivation. Belonging to this group, the study by Zhou et al. [51] is one of the few articles focusing on the relationship between the health status of caregivers and that of “care recipients”, e.g., spouses or older parents requiring care. This is important since informal caregivers often complain about their mental state (anxiety, depression, exhaustion, etc.). This study also explains how the income level of adult children influences caregiving decisions, since the likelihood of receiving assistance from one or more adult children appears to increase as their average income decreases. Butrica et al. [52] also focus on caregivers, although their article almost exclusively investigates the direct costs of parental or spousal caregiving. Carers are repeatedly described here as having few job opportunities and a lower percentage of asset growth.

Additionally, the article by Messer [53] can be cited as evidence that material deprivation among sick older people is occasionally partially self-imposed since they are ashamed to admit to their economic and health requirements. This is also one of the few qualitative studies we found, allowing us to observe how easily health costs may lead to a tense family environment.

Finally, in the seven papers that do not disclose a specific target in their objectives, older people emerge as the primary care recipient category, confirming that some literature tends to consider this category as a proxy for identifying care needs.

#### 3.1.2. The Material Dimension of Deprivation Attracts the Most Attention

Table 3 depicts the distribution of each deprivation dimension utilized by the reviewed articles. The data emphasize a traditional view of deprivation, as material wealth is the most frequently analyzed dimension (84.1% of publications), followed by health status (81%) and educational/social status (47%). Occupational status, social network ties, and marital status are mentioned in 35 cases, while the housing context is discussed in 30 of them. The level of social integration (16), work–life balance (4), perception of safety (3), and political participation (2) are the least cited dimensions.

Despite the trend to focus on material impoverishment, the definition of deprivation in 54 articles (85%) includes at least two dimensions. In ten of these papers (15.9%), the concept of deprivation comprises a low number of dimensions. Table 3 highlights that 44 studies (around 70%) applied a medium (42.9%) or high (27%) level of multidimensionality to the deprivation concept. From an overall analysis of the results presented in Table 3, it is possible to conclude that the material dimensions (e.g., material wealth, educational level, occupational level, and marital status) are preferred over others for describing deprivation. In most cases, multidimensional definitions of deprivation include at least one or more of them. These findings underline that social dimensions are viewed only as secondary or integrative components of the primary, largely material characteristics of the SED state of older people and their families.

#### 3.1.3. Little Room for a Two-Way Perspective of the Relationship between Healthcare Needs and SED

The emphasis placed on poverty and material deprivation by most studies impacts the design of the studies themselves. More often than not, the relationship between health and the deprivation of older people and families is examined by focusing on material impoverishment. Around 24% of publications included in the ScR (15 out of 63) discuss socioeconomic deprivation, while 60% (38 out of 63) examine the material impoverishment of people from the perspective of health conditions (Table 4). 

In particular, 24 articles (38.1%) discuss the financial impact by focusing on the financial burden as a result of chronic diseases and the subsequent healthcare consumption. In contrast, the relationship between people’s health and material deprivation is dealt with by 14 cases (22.2%). In ten papers (15.9%), the study objectives do not focused on the direct association between health and deprivation issues; instead, they only offer general reflections on the health and deprivation situations of older people and families, as is typical of review studies.

Forty-eight papers (76%) preferred to present a linear, one-way perspective of the relationship between older people’s health status and SE conditions. Ten articles (16%) adopted a two-way perspective to describe the relationship, therefore providing a more comprehensive view of this complex theme, and five publications (8%) approached the topic by discussing the indirect connection between the health status of older people and SE circumstances. Table 4 highlights that there is no favored route for observing the relationship: the number of studies (24) analyzing the health problems of older people as a factor impacting the SE situation corresponds to the number of investigations focusing in the opposite direction of the relationship.

#### 3.1.4. Paucity of Comparative Studies and Analyses of Primary Data

The ScR analysis enables the emergence of specific characteristics of geographical representativeness. More than 80% of the reviewed papers focus on a single country, while comparative studies are in the minority (17.5%).

Table 5 emphasizes that the vast majority of the research is undertaken in high- and middle-income countries; only one publication focuses on the issue in a low-income country. This is the article by Gabani et al. [54], which examines the percentage of Liberian households living below the so-called “poverty line” before and after taking out-of-pocket (OOP) health expenditures into account.

In relation to the considerably more regular availability of data for high-income countries, it is relevant to note that 84% of the papers reviewed are based on secondary data studies. In contrast, less than 5% are based on primary research studies (Table 6). 

This may be due to the greater availability of cross-sectional studies (42.9%) compared to longitudinal studies (20.6%).

### 3.2. Qualitative Contents Results

#### 3.2.1. The Main Findings on the Relationship between LTC Needs and SE Condition

A selection of articles underlines how the socioeconomic condition of individuals or families influences the health conditions of older people and their relative autonomy. The limits of the ADL are unevenly distributed by socioeconomic strata [55], and the mortality risk in poor older people is 1.71-times higher than in non-destitute elderly [56]. An in-depth reading of selected papers allows for identifying the socioeconomic factors affecting older adults’ health conditions. First, the socio-demographic characteristics are related to the level of education, gender, and marital status. Rising education levels correlate with better health or lower levels of disability in older people [24,57].

Furthermore, having a spouse counteracts the deterioration of health conditions in old age [58]. However, women are more frequently affected by disabilities than men [59]. Secondly, living in an area of social and material deprivation, e.g., a rural area and a disadvantaged neighborhood—is correlated with a higher disability rate, even though single socioeconomic factors can mitigate the risk of having poor health [60]. Agreeing with these results, Lima-Costa and colleagues [61] found that the provision of home care was inversely related to the socioeconomic gradient, identifying some particularities. The provision of formal care increased if education and family assets increased.

In contrast, informal care is less socioeconomically stratified but depends on the way of life of older people (e.g., living alone or living together). Indeed, families’ assets determine the ADL needs that will be covered [61]. In this regard, Aguila’s study [62] underlined how cash benefits seemed not to influence the familial caregiving asset: primary caregivers maintained their care-giving role and relative burden.

Due to the burden of care, the health of informal caregivers appears to deteriorate more rapidly and push them to retire 14 months earlier than those without caregivers [63,64], resulting in economic stress and a prolonged reduction in their assets [65,66]. Relatives are directly involved in financially supporting the coverage of the care costs of their older family members [49,67]. People receiving care, especially women, are at a high risk of impoverishment due to catastrophic healthcare spending [45,68,69,70].

Furthermore, a study conducted in South Korea in 2020 showed how the presence of disabled householders increased the risk of household multidimensional deprivation, with “poverty” being a concept inclusive of monetary and non-monetary dimensions. Park and Nam [71], the authors, went beyond income and asset measurements, identifying other crucial dimensions, such as subjective health condition, type and location of the house (e.g., basement floor; rooftop; non-residential building; or permanent/national rental apartment), and family and social relationships (i.e., satisfaction level).

In 2020, Del Pozo and colleagues [72] pointed out that cash benefit policies were ineffective in covering the need for care and protecting the family from ES deprivation. Furthermore, impoverished people could encounter difficulties accessing insurance or LTC schemes, promoting a circular causal process between SE deprivation, deteriorating health, and individual autonomy conditions [44].

#### 3.2.2. Suggestions for Future Studies

The selected studies widely expressed the need for future studies to focus on the causal relationship between the two phenomena studied [57,58,60]. Additionally, the studies invite future research to use better concepts relevant to an in-depth understanding of the relationships between LTC needs, the health condition of the elderly, and the risk of SE deprivation for individuals and families. First, socioeconomic deprivation enhances its multidimensional character [66,71] and the aspects of social exclusion that compose it. In particular, the studies examined encourage us to consider more the effects of (a) informal care on the loss of availability of working hours for carers [59,65] and on their retirement plans [63]; (b) living in a deprived neighborhood [60]; and (c) the living arrangements of dependent older people [56,61].

Secondly, the definition of informal carers should be open to neighbors and not only to cohabiting relatives [49,61,72]. Third, to measure the LTC needs, future studies should try to overcome the detection of health status, particularly if self-reported, preferring the use of ADLs limitations [57,72].

Moreover, the selected articles encouraged a better understanding of the relationship between SE conditions and (a) out-of-pocket expenditure burden [65,68,70], (b) implemented policies, and access to services and insurance schemes [62,72]. Lastly, longitudinal studies are auspicated to detect better the changes in long-term periods [55,59,60,61].

#### 3.2.3. Policies Implications

To reduce the financial and supply impacts related to the growing demand for care needs of LTC and welfare systems, some of the documents examined insisted on improving the prevention actions aimed at well-being and healthy aging in long-life courses [55,56,57,69]. Furthermore, several authors suggested paying greater attention to the welfare and social policies dedicated to individuals and families considered fragile and vulnerable due to their precarious health conditions [64] or their socioeconomic disadvantages [56,58,61]. Lima-Costa and colleagues [61] suggested considering dispositions and living conditions as relevant features of the profiles of deprived people to whom specific interventions should be dedicated. In particular, policymakers should pay attention to people living in rural areas and disadvantaged neighborhood contexts, where the level of unmet care and social security needs are usually higher than in urban and developed areas [24,59]

Considering the demographic trend and the reduction in household size, some papers push for reforming LTC and welfare systems and improving the quality of public formal service provision [63,65,67]. In this regard, Gabani and Guinness [54] suggested better integrating formal and informal assistance as a first step for improving interventions in support of informal caregivers. Supportive policies and interventions should be addressed to help working carers—mostly women—to maintain their jobs and balance their working and private lives [63,65,67,68], even in relation to cash benefit schemes to reduce the care burden [62]

## 4. Discussion

The analysis of the scientific literature demonstrated that there was an interest in the causal link between LTC needs and SED. However, it is often studied through a particular focus and unilateral way. Consequently, the ScR’s results highlight several gaps. The first relates to the definition of LTC needs. The widespread use of older population targets as proxies for the volume of LTC needs precludes a comprehensive analysis of the entire concept in all its complexity, including its composition in terms of the demand for health- and social care services [44,73].

Second, the use of older people as a proxy for identifying LTC needs contributes to an overrepresentation of care recipients in studies focusing on older people, even when the investigated problems are not strongly linked to the health- or social care received, and instead focus on the economic and social aspects.

However, the ScR identified some studies whose primary research target was caregivers and families, often defined by the “head” of the household. These two groups, however, are not jointly considered in the literature, indicating that the research usually prefers to focus on (and deal with) a single specific target rather than choosing a multiple-target population, which would more accurately reflect the complexity of most real-life LTC caregiving situations [6,73]

A third issue is that LTC needs are frequently defined in terms of health status or disability conditions, as opposed to ADL/IADL limitations, thus promoting a health-centered view of care needs. A similar simplification approach is also found about the multidimensional deprivation concept, which is greatly influenced by material and other easily measurable dimensions, resulting in the use of an idea of deprivation referring to the most traditional poverty and social inclusion definitions in most cases [74,75,76,77]. Consequently, when defining the socioeconomic conditions of families, the aspects connected to social life often remain undervalued. However, many studies identify them as pillars of informal carers and care recipients’ well-being and quality of life [78,79,80].

Nonetheless, SED and its core characteristics are gradually gaining prominence in policymakers’ formulations of the suggestions and recommendations for establishing LTC policies. Cash benefit schemes and support policies for working caregivers continue to be the main initiatives proposed to partially mitigate the effects of caregiving’s out-of-pocket financial burdens, even if their effectiveness is debated in the literature [27]. The more extensive availability of single-country studies and secondary data sources confirms that the scientific research in this field, in an effort to reduce the complexity of the triangle “LTC needs, health conditions of older people, and socioeconomic conditions”, has not yet found methodological and economically sustainable solutions that permit the gathering of more cross-national and primary data.

The multidimensional concept of SE is still lowly applied, even if the literature often pushes the need for this approach for future studies to use a more extensive idea of the target, including non-family members and informal carers. In the future decades, population aging will significantly accelerate in the countries of the global South [5], posing a new challenge. The lack of attention dedicated to date to low-income countries has created a significant gap in the evidence pertaining to these countries, thereby prohibiting an in-depth, urgently required analysis of the future sustainability of their developing welfare, health-, and social care systems [81]. Finally, the simplification strategy applied to many studies to lessen the complexity of the topic under investigation precludes an in-depth debate on some additional aspects. These include the understudied two-way relationship between LTC needs, supply, and the risk of socioeconomic deprivation; the marginal consideration of caregiving and SED’s social components in the majority of research; the widespread use of material poverty as a synonym for SED, which increases the risk of losing the numerous social exclusion aspects; and the lack of comparative or longitudinal studies. The multidimensional concept of SE is still lightly applied, even if the literature often encourages the need for this approach for future studies to use a more extensive concept of the target, including non-family members and informal carers. This suggestion supports the idea of the need for a widespread innovative systemic approach to care and welfare policies based on integration, and coordinated and preventive policies that want to respond to complex issues, taking into consideration the multidimensional aspects related to socioeconomic and care aspects. Table 7 summaries the main suggestion coming ftom this ScR.

Despite the wealth of information provided by this scoping review, some limitations should be considered when interpreting the results. These limitations are primarily attributable to the study’s exploratory objectives. In light of the dearth of literature recognizing ADL limitations in order to measure LTC needs, the set of keywords was broadened to include health conditions and disabilities. These two concepts do not always refer to dependent people. A similar search strategy was applied to the SED concept in conjunction with poverty and other material deprivation-related keywords, thus diminishing the selective power to offset their overrepresentation in the analyzed literature. The decision to use frequency distributions provided a user-friendly format for describing the results but precluded the detection of potential internal links among the selected variables. The exploratory perspective of this study also conditioned some methodological choices, excluding the use of a method for the quality analysis of the literature. A future systematic review on this topic will allow specific tools to evaluate the quality of the literature (e.g., MMAT). Moreover, in this scoping review, the descriptive exploration of territorial characteristics (n. studied countries and income level of countries) did not consider the differences related to the welfare states and care regimes that could influence the SED risk and health conditions of care recipients and their caregivers. To our knowledge, and despite these limitations, this study was the first to attempt to provide an overview of the literature examining the relationship between LTC needs and SED in both care recipients and caregiving families.

## 5. Conclusions

This ScR explored the relationship between LTC needs and the risk of socioeconomic deprivation for older people and their caregiving relatives. Detecting the interest in the literature on the issues, this ScR identified the main literature gaps and investigated the use of the multidimensional character of the SED concept. The relationship between LTC needs, the health status of older people, and the risk of socioeconomic deprivation for their families attracted the interest of specialized literature. Many studies adopted a simplification strategy to easily explore the high complexity of concepts and the crucial two-way relationship between LTC needs/supply and the risk of socioeconomic deprivation. This strategy did not allow for gaining in-depth knowledge of this relationship. Future studies should thoroughly analyze the causal relationship between the two concepts and uncover the underlying factors that characterize them. Systematic reviews and longitudinal studies should also be encouraged to foster a comprehensive understanding of the bidirectional influence between the two phenomena.

## Figures and Tables

**Figure 1 healthcare-11-02593-f001:**
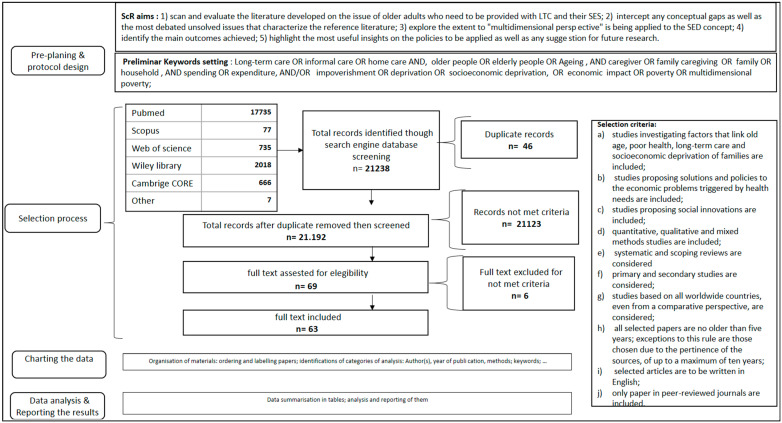
Flowchart of scoping review.

**Figure 2 healthcare-11-02593-f002:**
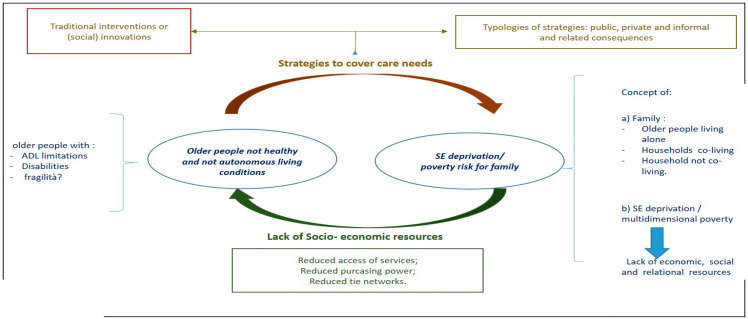
Conceptual framework of the relationship between LTC needs and socioeconomic deprivation (SED) risk.

**Figure 3 healthcare-11-02593-f003:**
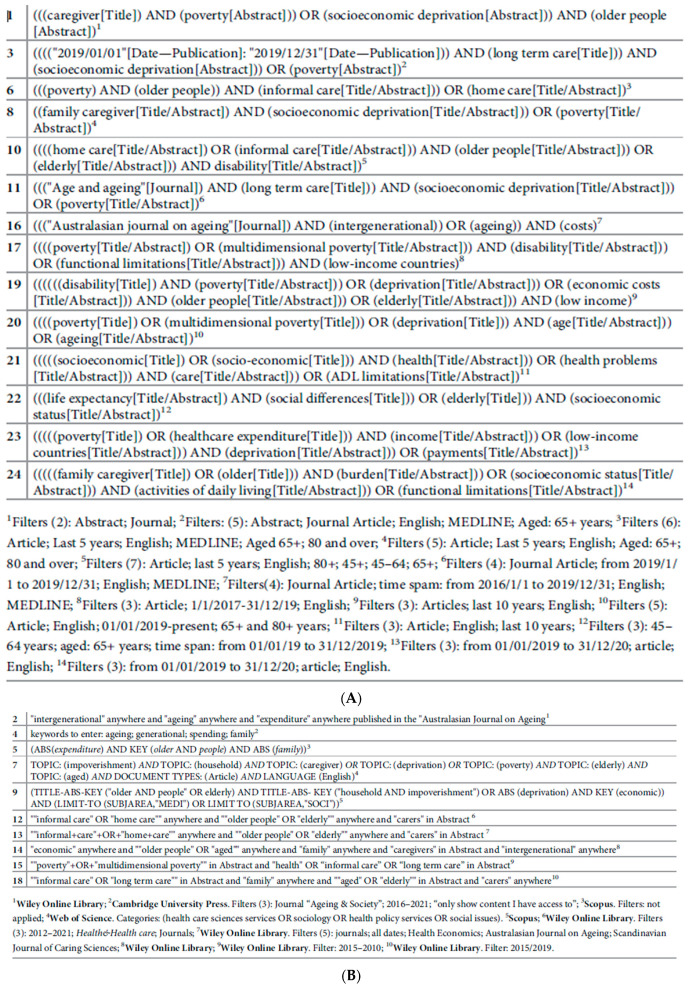
Search strings in Pubmed (**A**) and other search engines (Scopus, Web of Science, and Wiley Online Library) (**B**).

**Table 1 healthcare-11-02593-t001:** Study aims/research questions by selected analytical categories, analysis types, and table n. results.

	Table n.	Categories	Aims/Research Questions		
Quantitative analysis			**No. 1**	**No. 2**	**No. 3**
	2	Target of population	x	x	
	3	Distribution of deprivation dimensions ^1^			x
	3	Multidimensional deprivation level ^2^			x
	4	Focus (aims) of the study ^3^	x	x	x
	4	Perspective on the health–SED relationship ^4^	x	x	x
		Direction of health–SED relationship			
	5	Countries involved in the selected studies	x	x	
	5	Income level of the countries	x	x	
	6	Type of data (primary or secondary)	x	x	
Qualitative analysis	6	Typologies of design (longitudinal or cross-sectional studies)	x	x	x
	A2	Content results on the studied relationship	x	x	x
	A2	Suggestions for future studies	x	x	x

X: yes; ^1^: All the dimensions through which people—according to the authors of the selected articles—experienced deprivation (considering that this ScR aims to find out whether or not monetary and non-monetary dimensions were simultaneously included); ^2^: articles were scored on the basis of the number of dimensions considered; ^3^: purposes as contextualized and expressly argued by the authors (focus on title words, abstracts or, if present, dedicated paragraphs); ^4^: how the authors argued about the cause–effect relationship between the investigated factors, i.e., whether they used the one-way or the two-way concepts of the health–SED relationship (the former involves having a default setting whereby either health directly affects SED or SED directly affects health; the latter implies addressing the issue of bi-directionality).

**Table 2 healthcare-11-02593-t002:** Target population investigated.

Targets	n.	%
Older people	23	36.5
Households and/or heads of households	15	23.8
Caregivers	13	20.6
No specific target	7	11.1
Not applicable	5	7.9
Total	63	100

**Table 3 healthcare-11-02593-t003:** The concept of deprivation: dimensions and multidimensional deprivation levels.

Dimensions of Deprivation	n.	%
Material wealth (e.g., income; savings; assets)	53	84.1
Health status (self-reported health, health insurance coverage, and health services accessibility)	51	81
Education/social status	47	74.6
Occupational status	35	55.6
Social network ties	35	55.6
Marital status	35	55.6
Housing	30	47.6
Social integration level (e.g., presence or absence of barriers that prevent people from participating in society)	16	25.4
Work–life–leisure balance (e.g., caregiving burden in terms of lack of spare time)	4	6.3
Perceived safety	3	4.8
Political participation	2	3.2
Total	63	
**Multidimensional deprivation level (score 1–10)**	**n.**	**%**
High (range: 7–9)	17	27.0
Medium (range: 5–6)	27	42.9
Low (range: 2–4)	10	15.9
Not applicable	9	14.2
Total	63	100

**Table 4 healthcare-11-02593-t004:** Focus and direction of the investigated relationship between health, care needs, and SED.

Focus of the Study	n.	%
Relationship between health and socioeconomic deprivation (SED) factors	15	23.8
Relationship between health and material deprivation factors	14	22.2
Financial burden due to chronic conditions and healthcare consumption	24	38.1
General purposes	10	15.9
Total	63	100
**Direction of health–SED relationship**	**n.**	**%**
Health affects of socioeconomic conditions (health as an explanatory variable)	24	38
Socioeconomic conditions affect health (health as a dependent variable)	24	38
Two-way concept of the health–SED relationship (they mutually influence each other)	10	16
Other (i.e., indirect relationship)	5	8
Total	63	100

**Table 5 healthcare-11-02593-t005:** Territorial representativeness.

Number of Countries Involved in the Selected Studies	n.	%
One country (national or sub-national levels)	51	81
Two or more countries (cross-national research)	11	17.5
Not applicable (no country list)	1	1.5
Total	63	100
**Income level of the countries involved in the selected studies**	**n.**	**%**
Middle income	27	42.8
High income	32	50.8
HMICs	3	4.8
Low income	1	1.6
Total	63	100

**Table 6 healthcare-11-02593-t006:** Typologies of study: data and design.

Data Typology	n.	%
Secondary data analysis	53	84.1
Theoretical studies	7	11.1
Primary research studies	3	4.8
Total	63	100
**Type of design**	**n.**	**%**
Longitudinal	13	20.6
Cross-sectional	27	42.9
Others	23	36.5
Total	63	100

**Table 7 healthcare-11-02593-t007:** Summarized results and suggestions of study aims.

Aims	Main Results/Suggestions
To scan the literature on the topic of older adults who require LTC and their socioeconomic status	- Literature interests of the causal link between LTC needs and SED. - Several studies focus on the following questions, favoring the analysis of only one sense of the relationship: how does SED impact health conditions/disability, or how do health conditions affect SED conditions? - Specific issues are often studied to explain the relationship between LTC needs and SED (e.g., financial burden due to chronic conditions and healthcare consumption). - Literature underlines how living in socioeconomic deprived conditions and contexts affects the health status of older people, increasing the mortality rate of poor older people. - Education and gender are the socioeconomic characteristics that make a difference, even in the access and use of formal care services, while the provision of informal care does not show social stratification. - Health expenditure strongly influences the risk of poverty among older individuals and their families. In particular, the presence of disabled householders increases the risk of household multidimensional deprivation. - The literature underlines how supporting policies and cash benefits measures are ineffective in contrast to the adverse effects regarding SED risk and to support the health of older people and their caregivers.
To identify any conceptual gaps and the most debated unresolved issues in the literature	- Informal care as a focus of the studies is still under-explored. In particular, its open conception should be encouraged, including neighbors and/or friends. - The SED effects on working carers’ lives are still poorly studied. - The existing literature does not yet involve low-income countries in the studies of the issues. - Despite the existing literature, the burden of out-of-pocket expenditure on the SED risk of care recipients and caregivers must be studied in-depth. - The results underline how the impacts of policies and measures must be better studied. - Longitudinal and comparative studies are suggested.
To determine the extent to which the so-called “multidimensional perspective” is being applied to the SED concept	- The multidimensional perspective of SEDs is still hardly adopted in the literature, in particular, the material and social components (social participation and inclusion). - The results suggest that future studies should focus on the causal relationship between the two phenomena studied, based on an in-depth analysis of two concepts (LTC needs and SED) and their multidimensional characters.

## Data Availability

Not applicable.

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
