# Peer review of "The Impact of Long-Term Care Needs on the Socioeconomic Deprivation of Older People and Their Families: Results from Mixed-Methods Scoping Review"

_healthcare, 2023, doi:10.3390/healthcare11182593_

Round 1

Reviewer 1 Report

1.     There are some unclear symbols, or misspelling in lines 20, 22, 105, 292, 542 etc.

2.     Please list the full name of the SEreDIPE project.

3.     There are quantitative and qualitative analysis in the Table 1. But a quantitative approach was chosen in the line 138.

4.     The relationships between three research questions and the 3.1-3.7 results sections must be added to the text.

5.     There is incomplete paragraph among lines 312-314.

6.     The authors selected 12 categories to answer the research questions in Table 1. But the findings described in the section of results were 7 categories. These need to be clarified.

7.     The number labelled in the Table have to be consistent with the text, e.g., Table 3 or Table3.1, Table 3.2.

8.     There are three research questions (lines 134-138). The findings have to be described in the conclusions.

Author Response

Review 1

 The authors really give thanks for your suggestions, because they improve the quality of the paper.

1. There are some unclear symbols, or misspelling in lines 20, 22, 105, 292, 542 etc.

The author’s answers:  thanks for your suggestion, we checked all manuscript.

  1. Please list the full name of the SEreDIPE project.

The author’s answer: Done

  1. There are quantitative and qualitative analysis in the Table 1. But a quantitative approach was chosen in the line 138..

The author’s answer: Thanks for your suggestion; we revised the sentence to underline the mixed method approach better.

  1. The relationships between three research questions and the 3.1-3.7 results sections must be added to the text.

The author’s answer: Many thanks for your suggestion. According to your request, we decided to:

a) Reorganised the result section to underline quantitative and qualitative results better;

b) Improve table 1 related to variables and study aims, with table n. of results where the is described variable;

c) A table in Annex 2 has been added to summarise the qualitative data collection

d) A summarised table crossing study aims and results has been added to the discussion section (Table 7). The authors propose this colocation of the summarised table because it could be useful for the discussion and for not overput tables in the results section.

  1. There is incomplete paragraph among lines 312-314.

The author’s answer: Thanks for your suggestion; it is true; we cancelled the sentence because it was a typo. We apologise for these mistakes.

  1. The authors selected 12 categories to answer the research questions in Table 1. But the findings described in the section of results were 7 categories. These need to be clarified.

The author’s answer: As stated above, the results section and Table 1 have been reorganized to improve the clarity of the paper.Moreover, the result section has been revised.

  1. The number labelled in the Table have to be consistent with the text, e.g., Table 3 or Table3.1, Table 3.2.

The author’s answer: The number of tables was reorganised according to the text

  1. There are three research questions (lines 134-138). The findings have to be described in the conclusions

 The author’s answer: Thanks for your suggestion; we improved the conclusion section, including the aims of the study

  1. There is incomplete paragraph among lines 312-314.

The author’s answer:  Many thanks for your comment and attention. The sentence has been cancelled because it was a typo. 

Reviewer 2 Report

Report on “The Impact of Long-Term Care Needs on The Socioeconomic Deprivation of Older People and Their Families: Results from A Mixed Methods Scoping Review.”

Long-term care has become one of the biggest concerns in both the developed and developing worlds as human life expectations increase. This paper reviews peer-reviewed articles to examine the relationship between long-term care needs, health status of older people, and the risk of socio-economic deprivation for their households. This is a very important topic. I have some comments on the methods and findings below.

Method:

1. I think the selection criteria described from line 180 to line 191 are too broad and do not include any filter for the quality of the articles reviewed.  

2. The authors categorize the results by several dimensions: LTC defined by target (older people, caregivers, and households); the material dimensions of deprivation attract most attention; two-way perspective of the relationship between healthcare and SED; paucity of comparative studies and analysis if primary data, and the main findings on the relationship between LTC needs and SE conditions. I think these are strange ways to categorize the results; therefore, I don’t have a good picture after reading the paper. First, the results in developed and developing countries are mixed while their socio-economic conditions, institutions, and health insurance coverage are staggeringly different. For example, long-term care needs and financial pressure due to long-term care in the US differ greatly from those in Kenya and Liberia. So the results should be categorized by countries’ income levels. If the authors feel comfortable focusing on only one group of countries– high-income or low-and-middle-income countries, they can do so. Second, meta-analysis usually has some criteria to rank the quality of the paper. I don’t see that in the method sections. Do the authors give equal weights to all findings, regardless of the methods used (qualitative vs. quantitative, cross-sectional vs. longitudinal data, different methods of analysis…)

Results:

In the results section, the authors present five tables (Tables 2 – 6), but none of them actually report the findings of the articles reviewed. They report the dimensions used to categorize the articles. It would be more appropriate to put them in the Method section.

As I mentioned, there is no table reporting the findings of the articles reviewed. The findings of some articles are described in Section 3.5. in a not very organized way. After reading the entire article, I still don’t have an overall picture of the methods used and findings in the reviewed literature. The findings presented in the Results section were cited from less than 20 papers, while the number of articles reviewed is 63 (line 235). Why did the authors not summarize the findings of the other 40 articles? I suggest that the authors read and follow the structures commonly used in other meta-analysis papers to summarize and evaluate the findings of the literature of interest. There usually be a table in which authors list the findings of interest in one column, for example, “long-term care decreases the disposable income of the household,” and describe whether and how many of the papers reviewed confirm that and to what extent.  

Grammar and spelling need to be checked. Foe example, in the abstract, "analysis" should be "analysis".

Author Response

 Reviwer 2

Report on “The Impact of Long-Term Care Needs on The Socioeconomic Deprivation of Older People and Their Families: Results from A Mixed Methods Scoping Review.”

Long-term care has become one of the biggest concerns in both the developed and developing worlds as human life expectations increase. This paper reviews peer-reviewed articles to examine the relationship between long-term care needs, health status of older people, and the risk of socio-economic deprivation for their households. This is a very important topic. I have some comments on the methods and findings below.

The authors really give thanks for your suggestions, because they improve the quality of the paper.

Method:

  1. I think the selection criteria described from line 180 to line 191 are too broad and do not include any filter for the quality of the articles reviewed.  

The author answers: Many thanks for your suggestion.  The main aim of the scoping review is exploratory to identify gaps and general tips to address the new studies.  We consider the evaluation of the quality of literature to be a relevant topic more used in Systematic review. For this reason, I decided to include in the limitations section the lack of evaluation of the quality of literature (e.g., based on a maat method), suggesting to adopt it in a future literature review, in particular a Systematic review. 

  1. The authors categorize the results by several dimensions: LTC defined by target (older people, caregivers, and households); the material dimensions of deprivation attract the most attention; two-way perspective of the relationship between healthcare and SED; the paucity of comparative studies and analysis if primary data, and the main findings on the relationship between LTC needs and SE conditions. I think these are strange ways to categorize the results; therefore, I don’t have a good picture after reading the paper.

The author answers: Many thanks for your suggestion. We regret the easily readable results. We categorised the protocol paper (Martarelli et al., 2022) many times cited in the text. During the design process explained in the protocol paper ( Martarelli et al. 2022 ref.33 many times mentioned in the text),  based on the exploratory aims of the study Scoping review,  focusing the attention on if and how the multidimensional character of Sed concept and the mutual influence of relationship. The descriptive categories ( target and countries) are included in the results because of the descriptive categories of these items. However, we partially restructured the results section to help the readability of them.

  1. First, the results in developed and developing countries are mixed while their socio-economic conditions, institutions, and health insurance coverage are staggeringly different. For example, long-term care needs and financial pressure due to long-term care in the US differ greatly from those in Kenya and Liberia. So the results should be categorized by countries’ income levels. If the authors feel comfortable focusing on only one group of countries– high-income or low-and-middle-income countries, they can do so.

 The author answers: Many thanks for your suggestion. As already explained, the main aim of the scoping review is exploratory to identify gaps and general tips to address the new studies. Moreover, this scoping review does not consider the different levels of impact, but it wants to explore how literature studies this relationship. However, you have underlined a relevant topic on different impact related to SED conditions of people and welfare regimes, then we will put it in the limitations of the study.

Second, meta-analysis usually has some criteria to rank the quality of the paper. I don’t see that in the method sections. Do the authors give equal weights to all findings, regardless of the methods used (qualitative vs. quantitative, cross-sectional vs. longitudinal data, different methods of analysis…).

 The author answers: Many thanks for your suggestion; as explained above, we admit the lack of quality evaluation of literature, following the exploratory aims of ScR. We added this topic in the limitations section.

Results:

In the results section, the authors present five tables (Tables 2 – 6), but none of them actually report the findings of the articles reviewed. They report the dimensions used to categorize the articles. It would be more appropriate to put them in the Method section.

As I mentioned, there is no table reporting the findings of the articles reviewed. The findings of some articles are described in Section 3.5. in a not very organized way. After reading the entire article, I still don’t have an overall picture of the methods used and findings in the reviewed literature. The findings presented in the Results section were cited from less than 20 papers, while the number of articles reviewed is 63 (line 235). Why did the authors not summarize the findings of the other 40 articles? I suggest that the authors read and follow the structures commonly used in other meta-analysis papers to summarize and evaluate the findings of the literature of interest. There usually be a table in which authors list the findings of interest in one column, for example, “long-term care decreases the disposable income of the household,” and describe whether and how many of the papers reviewed confirm that and to what extent.  

articles. It would be more appropriate to put them in the Method section.

Many thanks for your suggestion,

 The qualitative analysis has been done in a selection of papers (including 22 papers) considered relevant by authors. This aspect of selection has been better detailed in the methods section. The reason for this choice to select papers for the qualitative part is related to the type of review ( Scoping review ) and his aims. Moreover, we don't want to produce a paper over full of tables. However, your suggestion pushed us to revise the results section partially, and  we added the following:

  1. a) Table in Annex 2 to summarise the qualitative data collection
  2. b) A summarised table crossing study aims and results has been added to the discussion section (Table 7). The authors propose this colocation of the summarised table because it could be useful for the discussion and for not overput tables in the results section.

Comments on the Quality of English Language

Grammar and spelling need to be checked. For example, in the abstract, "analysis" should be "analysis".

Thanks for your suggestion. We checked all the manuscript.

Reviewer 3 Report

Thank you for letting me review this manuscript. The manuscript deals with the association between LTC and SED in aged people and their care-givers and brings up a scoping review of articles tackling those issues. 

This is a very important manuscript shedding a light upon an almost unfamiliar and un-investigated relationships and as the authors mentioned - due to it's wide scope, the issue hasn't been broadly and in-depth investigated - which are this article targets. 

Some minor comments:

Line 33 - the authors mentions different aspects - such as?

Line 41 - the paragraph begins with the words "at the same time" but the relation between this current paragraph to the one before is unclear. Both paragraphs deals with different and unrelated issues that are not necessarily connected. 

The research model implies for a connection between LTC and SED but there are also people that are not deprived and use LTC, where are they within this model?

Following my last comment - Line 119 - does in all scoped countries and in all cases, the use or giving LTC leads to SED? what is happening in states with welfare regime?

Line 151 - There is a need in broad explanation regarding how authors choose the research key words. Same goes for line 158  regarding additional key words.

Author Response

Thank you for letting me review this manuscript. The manuscript deals with the association between LTC and SED in aged people and their care-givers and brings up a scoping review of articles tackling those issues. 

This is a very important manuscript shedding a light upon an almost unfamiliar and un-investigated relationships and as the authors mentioned - due to it's wide scope, the issue hasn't been broadly and in-depth investigated - which are this article targets. 

The authors really give thanks for your suggestions, because they improve the quality of the paper

Some minor comments:

Line 33 - the authors mentions different aspects - such as?

The author answers: Many thanks for your suggestion, the sentence has been improved, including examples of economic, social and human dimensions included in the multidimensional concept of poverty.

Line 41 - the paragraph begins with the words "at the same time" but the relation between this current paragraph to the one before is unclear. Both paragraphs deals with different and unrelated issues that are not necessarily connected. 

The author answers: Many thanks for your suggestion. The sentence has been separated from the previous one to underline better that another issue is starting to discuss

The research model implies for a connection between LTC and SED but there are also people that are not deprived and use LTC, where are they within this model?

The author answers: Many thanks for your suggestion; we adopt the conceptual framework used in a preview study (Casanova and Lillini 2021) where the LTC needs have an impact directly or indirectly (e.g. taxation)   on the income of the individuals.We added a sentence on it. Moreover, this scoping review does not consider the different levels of impact, but it wants to explore how literature studies this relationship. However, you have underlined a relevant topic on different impact related to  SED conditions of people and welfare regimes, then we will put it in the limitations of the study.

Line 151 - There is a need in broad explanation regarding how authors choose the research key words. Same goes for line 158 regarding additional key words.

The author answers: many thanks for your comment; we published a protocol paper where the selection of keywords process is strongly detailed; we cited the link with the protocol paper many times, e.g. “. As detailed in the protocol paper [33], the authors searched various databases using the keywords defined in the pre-planning phase that were strictly related to the above objective”.